# Current Status and Perspectives of Patient-Derived Models for Ewing’s Sarcoma

**DOI:** 10.3390/cancers12092520

**Published:** 2020-09-04

**Authors:** Tadashi Kondo

**Affiliations:** Division of Rare Cancer Research, National Cancer Center Research Institute, 5-1-1 Tsukiji, Chuo-ku, Tokyo 104-0045, Japan; takondo@ncc.go.jp; Tel.: +81-3-3542-2511

**Keywords:** Ewing’s sarcoma, patient-derived cancer model, cell line, organoid, xenograft, precision medicine, biomarker

## Abstract

**Simple Summary:**

A paucity of patient-derived cancer models hinders the development of novel therapeutic strategies in Ewing’s sarcoma. This review aimed to demonstrate the utility and possibility of popular patient-derived cancer models by overviewing the reported studies and to encourage the researchers to develop more models for Ewing’s sarcoma.

**Abstract:**

Patient-derived cancer models, including cell lines, organoids, and xenografts, are indispensable tools in cancer research. These models, which recapitulate molecular features of original tumors, allow studies on the biological significance of cancer-associated genes, antitumor effects of novel agents, and molecular mechanisms underlying clinical behaviors of tumors. Moreover, the predictive utility of patient-derived cancer models is expected to facilitate drug development and precision medicine. Ewing’s sarcoma is a highly aggressive mesenchymal tumor with a high metastasis rate. Previous studies demonstrated the utility of cell lines and xenografts in Ewing’s sarcoma research and clinical studies. However, the number of Ewing’s sarcoma models available from public biobanks is limited; this creates an obstacle for research on Ewing’s sarcoma. Novel Ewing’s sarcoma models are needed to establish their utility, further our understanding of the molecular mechanisms, and help develop effective therapeutic strategies. In this review, the current status of patient-derived cancer models is overviewed, and future prospects of model development are discussed from the perspective of Ewing’s sarcoma research. It should be of interest to researchers and clinicians who work on patient-derived cancer models.

## 1. Introduction

### Ewing’s Sarcoma and Patient-Derived Cancer Models

Ewing’s sarcoma is a highly aggressive mesenchymal tumor with a high metastasis rate in the lungs. Ewing’s sarcoma is the second most common bone tumor in children and adolescents after osteosarcoma, accounting for 3% of all childhood malignancies [1,2]. The genetic predispositions for Ewing’s sarcoma have not yet been established [3], and no preventive measures have been developed for Ewing’s sarcoma [4]. Ewing’s sarcoma is characterized by the presence of chromosome translocation, which causes the chimeric fusions; the t(11;22) chromosomal rearrangement causes the fusion of the *EWSR1* gene on chromosome 22 with the *FLI-1* gene on chromosome 11. The amino terminus of *EWSR1* fuses to the DNA-binding domain of *FLI1*, and a chimeric transcription factor *EWSRI-FLI-1* is generated [5]. The *EWSR1-FLI-1* fusion is observed in 85% of Ewing’s sarcoma cases [5,6]. *EWSR1-FLI1* plays a crucial role in the carcinogenesis and progression of Ewing’s sarcoma, regulating the expression of multiple genes [7,8]. Comprehensive gene and protein analyses have been conducted to investigate the molecular backgrounds of etiology and progression of Ewing’s sarcoma [9,10,11,12].

The treatment of Ewing’s sarcoma consists of a multimodal approach such as a combination of chemotherapy, surgery, and/or radiotherapy. Treatments for patients with localized Ewing’s sarcoma have significantly progressed over the past 30 years [13,14,15,16,17,18]. However, only 55% of patients receive appropriate chemotherapy [4], and there is much scope for improvement in treatment for patients with recurrence and resistance. Recently, novel therapeutic approaches for Ewing’s sarcoma have been developed based on the understanding of the molecular mechanisms of disease progression. For example, the therapeutic utility of a specific antibody [19], a small-molecule inhibitor [20], and their combination [21] targeting the insulin growth factor system has been examined in pre-clinical studies and clinical trials [22]. Numerous therapeutic agents, including monoclonal antibodies and small compounds, targeting the IGF1R pathway have been developed, and the results of clinical trials were well summarized in the previous review article [22].

Patient-derived cancer models have been critical tools for basic and pre-clinical research since the dawn of cancer research. The patient-derived cancer models, which faithfully capture the characters of original tumors, enable the functional studies for genomic aberrations, the evaluation of the antitumor effects of novel agents, and the validation of hypotheses of molecular mechanisms of diseases. Recent progress of genomics technologies generated a massive amount of clinically annotated genomics data [23,24], identifying mutation-defined subgroups, which could be associated with specific treatments [25,26,27,28]. The clarification of biological significance of genomic findings becomes more important. The clinical significance of genetic mutations depends on the cancer types [29,30,31], being influenced by the lineage and differentiation of tumor precursor cells [32]. Thus, we need cell-type-specific patient-derived cancer models to understand the genetic aberrations.

The major patient-derived cancer models are grouped into three types, namely cell lines, organoids, and xenografts, and they, individually, have unique features. Firstly, patient-derived cell lines enable high-throughput drug screening and genome-wide loss-of-function viability experiments [33,34,35,36,37]. Large-scale drug sensitivity screening using cell lines revealed the links between drug sensitivity and genetic alterations, leading to novel indications for anticancer agents [38,39] and discovery of candidate predictive biomarkers [40]. In addition, drug screening and gene function studies can be done at a low cost using cell lines. Numerous cell lines are deposited in public cell banks and are widely available for research use. Beside such advantages, monolayer cell lines lose their three-dimensional architecture and pivotal microenvironment components, such as stroma, vascular structure, and inflammatory cells. In addition, alterations in the genetic background, which affect drug responses, continuously occur during the propagation of cells because of deficient mechanisms of genome stabilization [41]. Moreover, the success rate of cell line establishment varies among cancer types, and tumor cells that are highly selected under survival pressure in tissue culture conditions are subjected to subsequent in vitro studies. Secondly, patient-derived organoids retain the three-dimensional structures of original tumor tissues; hence, they may represent the phenotypic features of original tumors better than the conventional tissue culture monolayer cell lines [42,43,44,45,46]. Furthermore, patient-derived organoids contain cancer stem cells or progenitor cells that are surrounded by microenvironment components. Moreover, the high take rate makes organoids more applicable for clinical practice than other cancer models. Organoids are expected to faithfully recapitulate the characters of original tumors, and have been used in studies of colorectal [47], gastric [48], pancreatic [49], breast [50], bladder [51], prostate [52], and kidney [53] cancers and glioblastoma [54]. However, organoids lack immune components, and organoids of only a limited type of cancers are available from public biobanks. Thirdly, patient-derived xenografts (PDXs) using tumor tissues have several unique advantages over other models. They retain microenvironmental components, such as stromal cells, immune cells, and vascular structure [55,56]. Moreover, PDXs contain the heterogeneous components of the original tumor tissue, suggesting their potentials for predictive modalities [57,58,59,60,61]. In addition to tumor tissues, circulating tumor cells (CTCs) are also promising sources of PDXs. CTCs were reported in patients with Ewing’s sarcoma, and their clinical utilities have been investigated using PDXs [62,63]. The biological behaviors and the metastatic potentials of CTCs can be evaluated by PDXs, and the anti-metastasis effects of drugs will be assessed using PDXs with CTCs. PDXs have been used for several decades in oncology, with the aim to develop clinical applications [64]. Besides their wide use, recent studies have indicated that the genomic contents of PDXs change during passaging in a way that does not resemble the progression of the original tumor; genomic evolution due to genetic instability is generally an issue in cancer models [65,66]. Moreover, human stromal components are gradually replaced with those of the experimental animals during passaging [67,68,69,70,71]. Because the genomic profile and tumor microenvironment affect the response to treatments, such alterations can reduce the utility of PDXs. These models have both pros and cons and complement the use of other patient-derived cancer models.

The purpose of this review article is to provide an overview of the current status of patient-derived models of Ewing’s sarcoma. The necessity of additional models for further insights into the etiology and progression of Ewing’s sarcoma is discussed. This review should be of interest to a wide range of researchers and clinicians who use patient-derived cancer models in their work.

## 2. Materials and Methods

The previous studies for Ewing’s sarcoma, where patient-derived cancer models were implemented, were investigated by searching the PubMed database (https://pubmed.ncbi.nlm.nih.gov/) using the keywords of Ewing’s sarcoma and the individual model names. After reviewing the search results, the papers with which the author was impressed in terms of novelty and sound results, and the papers demonstrating the possibilities of patient-derived cancer models, were selected. The process of paper selection depended on the author’s experience in the field of sarcoma research and patient-derived cancer models.

## 3. Results: Technical Variation of Patient-Derived Cancer Models Used for Research on Ewing’s Sarcoma

### 3.1. Cell Lines

Experiments using patient-derived cells provided novel insights into the molecular mechanisms of the etiology and progression of Ewing’s sarcoma. Several illustrative examples of the utility of cell lines in functional studies of the transcription factor EWSR1-FLI1 have been demonstrated. For example, Boulay et al. [72] reported that in in vitro tissue culture cells, EWSR1-FLI1 interacts with the Brahma homologue-related gene 1/ Brahma homologue-associated factor (BAF) chromatin remodeling complex and activates target gene transcription. Erkizan et al. [73] reported the interaction of EWSR1-FLI1 with multiple proteins in cell lines using mass spectrometry and demonstrated that RNA-helicase A could be a therapeutic target for Ewing’s sarcoma. Genome-wide gene knockdown assays using RNAi and cell lines revealed the function of EWSR1-FLI1. He et al. [74] reported that RNA interference (RNAi) screening targeting 6781 genes resulted in the identification of leucine-rich repeats and tryptophan-aspartic acid-repeat domain containing 1 (LRWD1) as a regulator of EWS-FLI1-driven cell viability. LRWD1-regulated transcriptional activity of EWSR1-FLI1 is associated with poor prognosis of patients with Ewing’s sarcoma. Another example of the utility of cell lines is the screening of oncology drugs. By screening a drug library, olaparib was identified to have remarkable in vitro activity against Ewing’s sarcoma [40,75], and these results were reproduced by other researchers [76,77,78]. The efficacy of olaparib was evaluated in phase II clinical trials [79]. Although the objective responses were not observed, the utility of patient-derived cancer models was proven in this case. Patient-derived cancer cells have also contributed to the study of the molecular mechanisms underlying disease progression. Ewing’s sarcoma tumor tissues include multiple types of tumor cells with different expression levels of EWSR1-FLI1. Franzetti et al. [80] followed up on these interesting observations by investigating the heterogeneous expression level of EWSR1-FLI1 using single cell reverse transcription polymerase chain reaction (RT-PCR) and imaging analysis, and demonstrated that EWSR1-FLI1 expression promoted lung metastasis [80]. Additional examples of the utility of cell lines in the study of Ewing’s sarcoma are too numerous to discuss here. Although some aspects of cellular physiology and heterogeneity may be inadequately retained in the cell lines, none of these studies could be performed if patient-derived cancer cells were not available for Ewing’s sarcoma.

Despite their substantial utility, cell lines for Ewing’s sarcoma are rarely available from public cell banks. According to a search of the world’s largest cell line database, Cellosaurus [81], there are only nine Ewing’s sarcoma cell lines deposited in cell banks. This is partly due to the number of patients with Ewing’s sarcoma being substantially small; the overall incidence of Ewing’s sarcoma averages 2.93 cases per million annually [14]. The number of cell lines needed to examine the antitumor effects of agents to generate enough data to promote further studies is unclear. In phase II clinical trials, where the efficacy of drugs is evaluated before the following phase III study, several dozens of patients need to be enrolled [82]. Although the absolute number of patients required for clinical trials with statistical power can be influenced by many factors, such as the expected efficacy of the drugs, prior probability of a favorable response, heterogeneity of the patient population, and the endpoint of the clinical trial, it is also desirable to include an equivalent number of cell lines from different patients in the preclinical study. In addition to the number of cell lines, cell lines with improved quality are needed. Most of the existing cell lines have inherent problems; generally, they have been passaged for many years, or even decades, and they are not adequately annotated with clinical and pathological data [83]. In addition, the discrepancy in results obtained from cell lines and those from clinical trials may be attributable to the instability of the genomic background of the cell lines and accumulation of mutations during multiple passaging [41]. Thus, after long-term culture, the cells should not be used for drug response assays. Recent studies indicated that short-term cultured cells present with features that are more similar to those of original tumors than long-term cultured cells, and thus are more useful for drug screening in both two- and three-dimensional culture conditions [37,44].

### 3.2. Organoids

Previous studies have indicated the potential utility of organoids for basic and clinical studies [42]. A literature search of PubMed using the keywords “sarcoma” and “organoid” revealed no reports on the study of sarcomas using organoids. This may be owing to the difficulty in establishing organoids using sarcoma tumor tissues, for unknown reasons. Alternatively, it may be simply because of the difficulty in obtaining adequate tumor tissues for experiments. If mesenchymal tumors are generally refractory to organoid formation, the investigation of mechanisms underlying such unique characteristics may lead to elucidation of novel biological aspects of organoids, or development of methodologies for establishing organoids.

### 3.3. PDXs

Previous studies employed and characterized PDXs of Ewing’s sarcoma. Nanni et al. [84] studied bone malignancies, including Ewing’s sarcoma. They implanted tumor tissues from 29 patients with Ewing’s sarcoma into mice and obtained PDXs in 7 of the 29 cases (24%). Gene expression analysis revealed that the expression level of all examined genes was not significantly different between PDX and tumor tissues. Patient-derived orthotopic xenografts (PDOXs) were also used in the study of Ewing’s sarcoma. The majority of subcutaneously implanted tumor tissues do not metastasize in immunocompromised mice; in contrast, the orthotopically implanted tumor tissues lead to metastasis [85]. Thus, PDOXs may better mimic the propensities of original tumors and may be more suitable to cancer research than the conventional PDXs. Miyake et al. [86] used PDOXs to demonstrate that treatment with regorafenib was effective against doxorubicin-resistant Ewing’s sarcoma. Miyake et al. [86] also demonstrated that the combination of temozolomide and irinote can inhibit the growth of a doxorubicin-resistant PDOX of recurrent Ewing’s sarcoma [87]. The antitumor effects of palbociclib, a CDK4/6 inhibitor, and listinib, an inhibitor of insulin receptor and insulin growth factor receptors, were examined using PDOXs of Ewing’s sarcoma with Fusion involved in t(12;16) in malignant liposarcoma-ETS-related gene fusion and cyclin-dependent kinase inhibitor 2A/B loss [88]. The antitumor effects of unique antitumor agents were also examined in PDOXs of Ewing’s sarcoma. Methionine addiction is a common feature of cancer, and methionine restriction causes cell cycle arrest. Thus, methionine is a therapeutic target in methionine-dependent cancers. Recombinant methionase (rMETase) can restrict the metabolism of methionine, and the antitumor effects of rMETase were reported in PDOXs of Ewing’s sarcoma [89]. *Salmonella* sp., a facultative anaerobe, preferentially replicates and colonizes within tumors when injected from a distal site, and this leads to tumor regression [90]. *Salmonella typhimurium* A1-R is a strain with high tumor colonization and antitumor efficacy [91]. Miyake et al. [92] demonstrated that combination treatments with rMETase and *Salmonella typhimurium* A1-R had significantly more prominent antitumor effects in PDOXs of Ewing’s sarcoma, compared with monotherapy with each agen. These reports demonstrated that PDXs and PDOXs can facilitate drug development and serve as tools of pre-clinical study.

The feasibility of using PDXs for precision medicine was suggested in a previous study. Stebbing et al. [93] reported how they used PDXs for precision medicine in Ewing’s sarcoma. In their study, the tumor tissues from three patients with Ewing’s sarcoma were examined, and data were obtained from two of the three. The effects of chemotherapy were compared between the PDXs and the tumors from which they established PDX in one case. A combination of docetaxel, gemcitabine, and bevacizumab proved effective in the PDXs. Based on the results of this experiment, the patient was treated with the same combination of anticancer drugs. A partial response was observed after two cycles of treatment, and after an additional five treatment cycles, the beneficial effects continued for more than six months [93]. These observations suggested how we can use PDXs for drug development and precision cancer medicine. Despite these promising observations and expectations for clinical applications, according to a search of the PDX database PDXFinder (https://www.pdxfinder.org/), there are only five PDXs of Ewing’s sarcoma that are currently available from public biobanks. Clearly, more PDXs are needed for preclinical studies.

## 4. Conclusions

### Future Perspectives of Patient-Derived Ewing’s Sarcoma Models

This review presented several examples of the considerable possibilities for using patient-derived cancer models. The examples demonstrated in this review showed that adequate models can facilitate research considerably. Although the patient-derived cancer models have versatile utilities, each model has its own pros and cons, and we should use different models depending on the situation. Because of the limited number of patients, such models are seriously difficult to obtain in Ewing’s sarcoma. In other words, the delay in effective treatments in patients with Ewing’s sarcoma is partially attributable to this paucity of adequate cancer models. Indeed, since this disease was defined about one century ago, only a limited number and variation of models were developed. Tumor tissues for Ewing’s sarcoma research are limited owing to the exceptionally low incidence and prevalence of this type of cancer, and the researchers who can access such models have significant advantages over other researchers who cannot, and at the same time, they have a responsibility to use them in a proper way. The establishment of cancer models should not be performed by researchers in a competitive way; rather, cancer models should be shared as public resources and made available to subsequent generations.

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
