# Peer review of "Current Status and Perspectives of Patient-Derived Models for Ewing’s Sarcoma"

_cancers, 2020, doi:10.3390/cancers12092520_

Round 1

Reviewer 1 Report

The review manuscript entitled “Current Status and Perspectives of Patient-Derived 2 Models for Ewing’s Sarcoma” is very interesting. It summarizes the common models for the study of important genes and pathways implicated to Ewing’s sarcoma. Some comments that could help to improve this review are

  1. It would be interesting if the author will further analyze the treatment with the antibody mentioned in line 57 (reference num. 20).
  2. In our days CDX models (xenografts from patients’ CTCs) could also be used for the study of signal transduction pathways and treatments. This fact could also be mentioned in the introduction where author summarizes the common methods that can be used for the study of different type of cancers.
  3. There are also studies (Benini et al 2018, Hayashi et al 2017 etc) regarding the presence of Circulating Tumor Cells (CTCs) in patients with Ewing’s sarcoma. These studies can give interesting information for the understanding of metastatic spread in sarcomas and they can be mentioned in the manuscript.

Author Response

Comment 1

It would be interesting if the author will further analyze the treatment with the antibody mentioned in line 57 (reference num. 20).

Response to comment 1

Accordingly, the review article, which summarized the results of clinical trials, was referred in the revised manuscript.

Comments 2 and 3

In our days CDX models (xenografts from patients’ CTCs) could also be used for the study of signal transduction pathways and treatments. This fact could also be mentioned in the introduction where author summarizes the common methods that can be used for the study of different type of cancers.

There are also studies (Benini et al 2018, Hayashi et al 2017 etc) regarding the presence of Circulating Tumor Cells (CTCs) in patients with Ewing’s sarcoma. These studies can give interesting information for the understanding of metastatic spread in sarcomas and they can be mentioned in the manuscript.

Response to comments 2 and 3

Accordingly, the presence of CTCs in patients with Ewing’s sarcoma is mentioned in the revised manuscript.

Reviewer 2 Report

This is a review that summarizes the current status of patient-derived models of Ewing’s sarcoma and it gives further insight in patient derived cancer models. However, the presentation of the text needs improvement. I would suggest a i) "method" section where the search strategy for this review will be analysed, including the databases that were searched, the search terms, any inclusion/exclusion criteria, dealing with duplicates, if any, time period of search. ii) a clearly defined "result" section where the results of the search should be presented, as well as a summary of the results presented in a table and iii) a "discussion" section.

Author Response

Comment

 I would suggest a i) "method" section where the search strategy for this review will be analysed, including the databases that were searched, the search terms, any inclusion/exclusion criteria, dealing with duplicates, if any, time period of search. ii) a clearly defined "result" section where the results of the search should be presented, as well as a summary of the results presented in a table and iii) a "discussion" section.

Response

According to the suggestion, the manuscript is formatted with subtitles. However, the summary of results in the table is difficult to draw, because of the contents of this article.